# MMRN: A Multi-scale Multi-task Residual Network for Seasonal Climate Forecasting

## Abstract

Seasonal climate forecasting has high social and economic benefits in many fields. Although spatial multi-scale modeling of the coupled ocean-atmosphere climate system is a promising direction for seasonal forecasting, two key challenges limit it further development: (1) Feature entanglement across scales and variables not only hinders the disentangling and learning of multi-scale features but also introduces heterogeneous interference. (2) Hierarchical error propagation causes errors to accumulate at the final forecasting scale, compromising the forecast skill. To this end, we propose Multi-scale Multi-task Residual Network (MMRN) for seasonal climate forecasting. Specifically, to mitigate the feature entanglement, a group-wise variables embedder module is first introduced to independently embed the ocean and atmosphere variables, then a residual multi-scale feature extractor module is designed to extract disentangled multi-scale features. These features are then modeled to generate forecast at each scale. In addition, a multi-task learning mechanism is employed in the learning process by imposing scale-specific supervision to curb the hierarchical error propagation. Extensive experiments on the reanalysis dataset demonstrate that MMRN achieves state-of-the-art performance compared to the advanced deep-learning based models. Code is available at https://anonymous.4open.science/r/MMRN-4FB5/.

## 1 Introduction

Seasonal climate forecasting, which can predict meteorological variables 3 to 6 months in advance, is crucial for energy management, agricultural planning, and preparedness for extreme weather events (e.g., droughts and floods) (Bruno Soares et al., 2018; Han et al., 2024; Xu et al., 2025). Accurate seasonal climate forecasting relies on capturing boundary conditions at the Earth's surface (e.g., sea surface temperature) and modeling the evolving internal variability of the coupled ocean-atmosphere climate system, which is different from short-term weather forecasting (up to 3 days) that mostly depends on atmospheric initial conditions (Chakraborty & Veeresha, 2024).

Traditional seasonal climate forecasting is based on dynamical models (Xing et al., 2024), which simulate dynamics in the climate system by solving complex thermodynamic and fluid dynamics equations that describe how the climate evolves. Despite the notable progress achieved by dynamical models in seasonal climate forecasting (Jeong et al., 2022; He et al., 2022), these models generally require substantial computational resources for solving complex equations (Balaji et al., 2017), exhibit model deviations due to incomplete physical process modeling, and fail to make use of the rapidly growing amount of observational data (Guo et al., 2025).

Recently, benefiting from the powerful capabilities in non-linear modeling and big data processing (Chen et al., 2023a; Singh & Tyagi, 2024), deep learning models have been increasingly explored for seasonal climate forecasting to overcome the limitations of dynamical models (Olivetti & Messori, 2024). Within these studies, spatial multi-scale modeling of the climate system is a promising direction, as the climate system exhibits a distinct spatial multi-scale characteristic, spanning from large-scale climate variability (e.g., El Niño-Southern Oscillation, ENSO) (McPhaden, 2002) to small-scale climate variability (e.g., local orographic precipitation) (Roe, 2005). Existing spatial multi-scale models are generally based on the assumption that the original scale corresponds to the smallest scale (Ronneberger et al., 2015). They typically follow a four-step paradigm: multi-scale feature extraction, feature interaction modeling, cross-scale information aggregation, and fi-

nally forecasting at the original scale (i.e., smallest scale) (Chen et al., 2023c). However, we argue that two challenges hinder the spatial multi-scale modeling of the coupled ocean-atmosphere climate system for accurate seasonal climate forecasting.

(1) **Feature Entanglement**. The feature entanglement manifests in two distinct aspects. Firstly, *cross-scale feature entanglement* arises from the inherent coupling and co-evolution of the large-scale and small-scale climate patterns in the climate system, e.g., the coupled evolution between ENSO and local convection. This entanglement compels a model to represent multi-scale features in a shared feature space, thereby challenging the model's ability to disentangle and learn multi-scale features (Locatello et al., 2019). The challenge is particularly severe for entangled small-scale features, as the inherent spectral bias of deep learning models causes the learning process to be dominated by low-frequency and large-scale features, while high-frequency and small-scale features are neglected (Rahaman et al., 2019). Secondly, *cross-variable feature entanglement* stems from the physical coupling among different ocean and atmosphere variables (Rogers, 1995). This entanglement results in heterogeneity interference due to the fundamentally different physical characteristics of ocean and atmosphere variables (Baltrušaitis et al., 2018). Even worse, the interference propagates across all scales during the multi-scale feature extraction and finally gets amplified.

(2) **Hierarchical Error Propagation**. The cross-scale information aggregation in the spatial multi-scale modeling is achieved by creating cross-scale connections to fuse information from different scales (Tang et al., 2023). Unfortunately, these connections also act as direct pathways, which allow errors originating at one scale to propagate freely throughout the hierarchical structure (Jiang et al., 2024). Ultimately, these errors converge at the smallest scale, as this is where the final forecasting occurs. This process inevitably compromises the forecast skill of seasonal climate forecasting. Even worse, the hierarchical error propagation further exacerbates the neglect of small-scale features induced by the spectral bias, as the errors propagated from larger scales are dominant in the training process (Li et al., 2022).

Motivated by the above, we propose MMRN, a Multi-scale Multi-task Residual Network for seasonal climate forecasting. To the best of our knowledge, MMRN is the first work that applies residual spatial multi-scale modeling and multi-task learning mechanism to the coupled ocean-atmosphere climate system for seasonal climate forecasting. The main contributions are summarized as follows:

- We propose a residual multi-scale feature extractor (ResMFE) module. Starting from the largest scale, it iteratively subtracts the extracted features of the current scale from the input, generating residual input for the next smaller scale. The module can extract pure features at each scale that are disentangled from larger scale features. In addition, we introduce a group-wise variable embedder (GVE) module to embed ocean and atmosphere variables independently, which can mitigate interference from cross-variable feature entanglement.

- We design a multi-task learning mechanism by defining a specific forecasting task for each scale to optimize the overall learning process. This mechanism can not only directly curb the hierarchical error propagation, but also provide explicit supervision for small-scale patterns, thereby counteracting the neglect of small-scale patterns induced by spectral bias.

- Extensive experimental results demonstrate that MMRN achieves the state-of-the-art (SOTA) performance, with an average reduction of 6.04% in RMSE and improvement of 9.12% in ACC on 8 target variables compared to the best baseline.

## 2 RELATED WORK

Deep learning models have significantly advanced weather and climate forecasting by effectively modeling the complex internal dynamics of the climate system (Olivetti & Messori, 2024; Chen et al., 2023b; Zhang et al., 2014). These models can be broadly categorized by their architectures. For example, based on Vision Transformer (ViT) (Dosovitskiy et al., 2020), ClimaX (Nguyen et al., 2023), Pangu (Bi et al., 2023), and CirT (Liu et al., 2025) treat the patches of the climate grid as tokens and utilize the self-attention mechanism to model spatial dependencies within the climate system. However, they all overlook the inherent spatial multi-scale characteristic in the climate system and adopt single-scale modeling, which limits their forecasting performance. Based on the Graph Neural Network (GNN) (Zhou et al., 2020), GraphCast (Lam et al., 2023) and OneForecast (Gao et al., 2025) treat the climate system as a multi-scale spherical mesh. By defining grids as

nodes and their connections as edges, these methods perform message passing on graphs of different spatial scales to model the internal correlations of the climate system (Gilmer et al., 2017). Nevertheless, all of them fail to both extract disentangled multi-scale features and mitigate hierarchical error propagation, which are two challenges inherent to most existing spatial multi-scale modeling frameworks. To address these challenges, we apply the residual spatial multi-scale modeling with multi-task learning mechanism to the climate system. Specifically, we employ a ResMFE module and a GVE module to extract disentangled multi-scale features. In addition, we design a multi-task learning mechanism to impose supervision of each scale, thereby curbing the hierarchical error propagation.

## 3 PRELIMINARY

**Problem Definition.** Given the initial monthly climate state $X_t \in \mathbb{R}^{C \times H \times W}$ at time step $t$, where $C$ represents the total number of ocean and atmosphere variables, $H$ and $W$ are the height and width of the global latitude-longitude grid, respectively. The task of the seasonal climate forecasting is to predict future climate state $X_{t+T} \in \mathbb{R}^{C \times H \times W}$ at a target lead time of $T$ months, which can be formulated as follows:

$$\widehat{X}_{t+1} = \mathcal{F}(X_t; \theta), \tag{1}$$

where $\mathcal{F}$ is the seasonal climate forecasting model and $\theta$ denotes the learnable parameters of the model $\mathcal{F}$. For a longer lead time $T > 1$, we iteratively feed the previous forecast as the next input until the target lead time is reached, which can be formulated as follows:

$$\widehat{X}_{t+i+1} = \mathcal{F}\left(\widehat{X}_{t+i}; \theta\right), \quad \text{for } i = 1, 2, \ldots, T-1. \tag{2}$$

This auto-regressive forecasting strategy allows us to forecast future climate state at any desired lead time using only one trained model.

**Training Strategy.**

**(1) Pre-training** In the first stage, we pre-train MMRN on the datasets from the Coupled Model Intercomparison Project Phase 6 (CMIP6) (Eyring et al., 2016), which consist of climate simulations of the coupled ocean-atmosphere climate system from a diverse ensemble of physics-based models. By learning from the vast and diverse physically consistent simulation data, the model is guided to capture the general variability within the climate system and acquire crucial underlying physical knowledge (Nguyen et al., 2023).

**(2) Fine-tuning** In the second stage, we fine-tune MMRN on a coupled ocean-atmosphere dataset, which is constructed by spatially and temporally aligning ocean variables from the Ocean Reanalysis System 5 (ORAS5) (Matsueda & Palmer, 2018) with atmosphere variables from the fifth generation of ECMWF Reanalysis Data (ERA5). Both are observation-constrained reanalyses, widely regarded as the best available estimates of the historical climate state (Hersbach et al., 2020). The fine-tuning process adapts the general physical knowledge learned during pre-training phase to the specific climate patterns of the real world, thereby bridging the gap between climate simulations and observations.

## 4 MMRN

The core of MMRN is the synergistic application of residual spatial multi-scale modeling and multi-task learning mechanism to the coupled ocean-atmosphere climate system for accurate seasonal climate forecasting. In doing so, given the initial climate state, we tackle the feature entanglement by first applying a GVE module to embed ocean and atmosphere variables independently, and then leveraging two parallel ResMFE modules to extract disentangled multi-scale ocean and atmosphere features, respectively. These features are then fed into a series of ViT-based predictors to model the internal correlations of the climate system and generate a forecast at each scale. Finally, to compensate for the information loss caused by the ResMFE module, we aggregate these forecasts to form the future climate state at the original scale. Notably, the overall learning process is optimized by the multi-task learning mechanism, which defines a separate forecasting task for each scale, thereby curbing the hierarchical error propagation. The overall framework of MMRN is illustrated in Fig. 1.

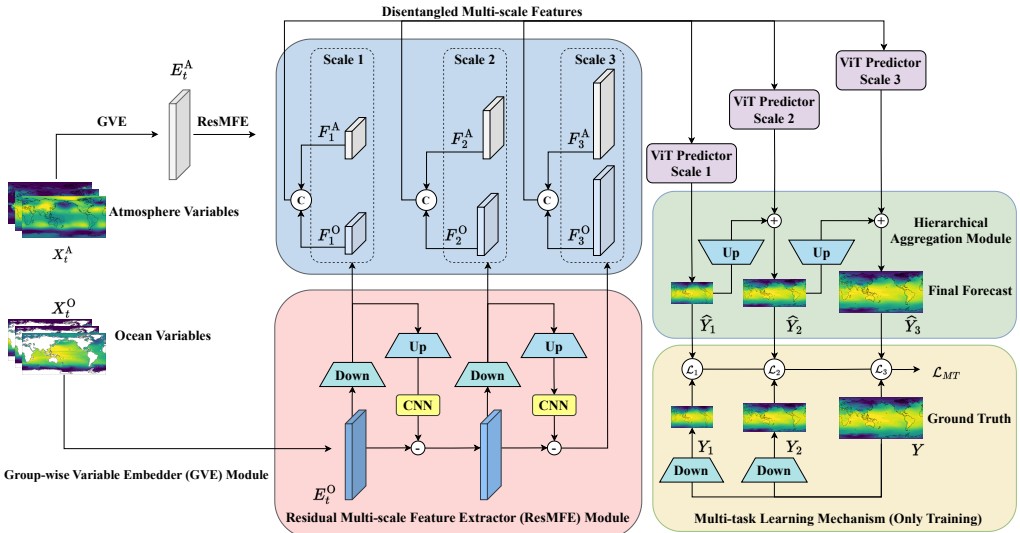

Figure 1: The framework of MMRN.

**Group-wise Variable Embedder (GVE) Module** The GVE module is designed to mitigate the heterogeneity interference induced by cross-variable feature entanglement in the coupled ocean-atmosphere climate system at the root, which can provide disentangled ocean and atmosphere embeddings for the subsequent modules, especially the ResMFE module.

Suppose $X_t \in \mathbb{R}^{V \times H \times W}$ denotes the input climate state, we first partition the input along the variable dimension into two groups: the ocean variables $X_t^O \in \mathbb{R}^{V_O \times H \times W}$ and the atmosphere variables $X_t^A \in \mathbb{R}^{V_A \times H \times W}$, where $V_O$ and $V_A$ denote the number of ocean and atmosphere variables, respectively. After that, the ocean and atmosphere variables are fed into two independent embedding networks, which can be formulated as follows:

$$
\begin{aligned}
E_t^O &= \text{Embed}_O(X_t^O; \theta_O), \\
E_t^A &= \text{Embed}_A(X_t^A; \theta_A),
\end{aligned}
\tag{3}
$$

where $\text{Embed}_O$ and $\text{Embed}_A$ are two independent embedding networks (e.g., linear layer) with their own learnable parameters $\theta_O$ and $\theta_A$, respectively. $E_t^O \in \mathbb{R}^{D \times H \times W}$ and $E_t^A \in \mathbb{R}^{D \times H \times W}$ are disentangled ocean and atmosphere embeddings, respectively. In addition, the treatment of undefined values for ocean variables over land is described in Appendix A.1.

**Residual Multi-scale Feature Extractor (ResMFE) Module** After obtaining the ocean and atmosphere embeddings, a conventional approach to extract multi-scale features is to iteratively downsample the input embedding from small-scale to large-scale (Shang et al., 2024a;b). However, we argue that this approach suffers from the inherent cross-scale feature entanglement, especially small-scale features entangled by large-scale features, as the dominance of large-scale features in the learning process hinders the effective learning of small-scale features.

To address this issue, we propose the ResMFE module to extract pure features at each scale that are disentangled from larger scale features in a large-scale to small-scale order. At each scale, the ResMFE module extracts scale-specific features by downsampling the current input. These features are then upsampled to the original resolution and subtracted from the current input, creating a residual that is stripped of larger scale information and serves as the input for the next iteration. Suppose $E \in \mathbb{R}^{D \times H \times W}$ denotes the input embedding of the ResMFE, this iterative procedure can be formulated as follows:

$$
\begin{aligned}
F_s &= \text{Down}(E_s), \\
E_{s+1} &= E_s - \phi_s(\text{Up}(F_s)),
\end{aligned}
\tag{4}
$$

where the iteration is initialized with the base case $E_0 = E$. $s = 0, 1, \ldots, S-1$ denotes scale index and $S$ is the total number of scales. $E_s \in \mathbb{R}^{D \times H \times W}$ denotes the residual input embedding at scale

$s$. $F_s \in \mathbb{R}^{D \times H_s \times W_s}$ denotes the scale-specific features at scale $s$. $H_s$ and $W_s$ are the height and width at scale $s$. In particular, the smallest-scale preserves the original resolution, i.e., $H_{S-1} = H$ and $W_{S-1} = W$. Down and Up denote spatial resampling operators (e.g., bicubic interpolation) for downsampling and upsampling, respectively. $\phi_s$ denotes a lightweight convolutional block at scale $s$ that is employed to address the information loss in downsampling and upsampling.

By iteratively extracting features from a residual input from which larger scale information has been subtracted, the ResMFE module can explicitly extract disentangled multi-scale features, thereby mitigating the cross-scale feature entanglement. Notably, the ocean and atmosphere embeddings are processed independently by two parallel ResMFE modules to extract ocean and atmosphere multi-scale features, respectively.

**Vision Transformer (ViT)-based Predictor** After obtaining the disentangled ocean and atmosphere multi-scale features, we introduce a series of ViT-based predictors. Each predictor is designed to model the spatial correlations within the coupled ocean-atmosphere climate system and generate the forecast at each scale, following a four-step process: early fusion, tokenization, Transformer encoding, and forecasting. For each scale, we first employ an early fusion strategy by concatenating the ocean features $F_s^O \in \mathbb{R}^{D \times H_s \times W_s}$ and atmosphere features $F_s^A \in \mathbb{R}^{D \times H_s \times W_s}$ along the channel dimension, yielding coupled ocean-atmosphere features $F_s^C \in \mathbb{R}^{2D \times H_s \times W_s}$. The tokenization begins by dividing these coupled features into $N_s$ patches, where $N_s = (H_s W_s / p_s^2)$ is the number of patches and $p_s$ is the patch size. Each patch is then embedded into $D$ dimensions to obtain a sequence of token embeddings $T_s \in \mathbb{R}^{N_s \times D}$. For Transformer encoding, these token embeddings are first added with learnable position embeddings $P_s \in \mathbb{R}^{N_s \times D}$ to retain positional information. Then the resulting token embeddings are fed into $L$-layer Transformer encoders to model the spatial correlations within the climate system. Finally, we employ a prediction head that takes the output $T_s^L \in \mathbb{R}^{N_s \times D}$ from $L$-layer Transformer encoders as input and generates forecast $\widetilde{Y}_s \in \mathbb{R}^{V \times H_s \times W_s}$ at scale $s$. The prediction head consists of two linear layers and a convolution layer.

**Hierarchical Aggregation Module** Notably, the ResMFE module extracts small-scale features from residual input where larger scale features have been subtracted, rendering the small-scale forecasts inherently devoid of the foundational large-scale patterns. To address this issue, we employ a hierarchical aggregation module that iteratively aggregates forecasts from large-scale to small-scale. Specifically, the forecast at each scale is enhanced by integrating the raw forecast of the current scale with the upsampled aggregated forecast from the previous larger scale, which can be formulated as follows:

$$\widehat{Y}_s = \mathrm{AGG}(\widetilde{Y}_s, \mathrm{Up}(\widehat{Y}_{s-1})), \quad \text{for } s = 1, \ldots, S-1, \tag{5}$$

where the iteration is initialized with the base case $\widehat{Y}_0 = \widetilde{Y}_0$. $\widetilde{Y}_s, \widehat{Y}_s \in \mathbb{R}^{V \times H_s \times W_s}$ are the raw forecast and the aggregated forecast at scale $s$, respectively. AGG denotes the aggregation function (e.g., element-wise addition). Up denotes a spatial resampling operator used to align resolution. The final forecast for the future climate state is given by $\widehat{Y}_{S-1} \in \mathbb{R}^{V \times H \times W}$.

**Multi-task Learning Mechanism** In the single-task learning mechanism employed by most existing works, the optimization is driven solely by the discrepancy between the final forecast and the ground truth. However, we argue that this approach is fundamentally flawed for multi-scale architectures like ours. By neglecting the supervision on other scales, the single-task learning mechanism fails to mitigate the inherent hierarchical error propagation, allowing errors to accumulate at the original scale and amplifying the neglect of small-scale features induced by spectral bias. To this end, we introduce a multi-task learning mechanism by defining a specific forecasting task to directly impose supervision at each scale.

During the training phase, given the ground truth of future climate state $Y \in \mathbb{R}^{V \times H \times W}$, we construct a forecasting target at each scale by downsampling the ground truth using average pooling, which can be formulated as follows:

$$Y_s = \mathrm{AvgPool}_s(Y), \quad \text{for } s = 0, \ldots, S-2, \tag{6}$$

where $Y_s \in \mathbb{R}^{V \times H_s \times W_s}$ is the forecasting target at scale $s$. $\mathrm{AvgPool}_s$ denotes an average pooling operation with the appropriate parameters to downsample the ground truth $Y$ to the spatial resolution $H_s \times W_s$ at scale $s$. Specifically, the forecasting target at the smallest scale is the ground truth, i.e., $Y_{S-1} = Y$. The multi-task loss $\mathcal{L}_{MT}$ is subsequently defined as the weighted sum of the latitude-

weighted mean squared error loss $\mathcal{L}_{LM}$ across all scales, which can be formulated as follows:

$$\mathcal{L}_{MT} = \sum_{s=0}^{S-1} \lambda_s \mathcal{L}_{\text{LM}}(\widehat{Y}_s, Y_s),$$

$$\mathcal{L}_{\text{LM}}(\widehat{Y}_s, Y_s) = \frac{1}{V \times H_s \times W_s} \sum_{v=1}^{V} \sum_{i=1}^{H_s} \sum_{j=1}^{W_s} L_s(i) \left( \widehat{Y}_s^{v,i,j} - Y_s^{v,i,j} \right)^2, \quad (7)$$

$$L_s(i) = \frac{\cos(\phi_i)}{\frac{1}{H_s} \sum_{i'=1}^{H_s} \cos(\phi_{i'})},$$

where $\widehat{Y}_s$ and $Y_s$ denote the aggregated forecast and forecasting target at scale $s$, respectively. $\lambda_s$ denotes the weight that balances the contribution of the loss at each scale. $L_s(i)$ denotes the latitude weighting factor of the $i$th latitude $\phi_i$ at scale $s$. By imposing direct supervision for each scale, the multi-task learning mechanism enforces accurate forecasting at each scale, thereby curbing hierarchical error propagation.

## 5 EXPERIMENTS

**Datasets** We pre-train MMRN on 9 different datasets from CMIP6 and fine-tune MMRN on a coupled ocean-atmosphere dataset, which is constructed by aligning datasets from ERA5 and ORAS5. For all the data, the temporal resolution is month while the spatial resolution is $5.625° \times 5.625°$ with a $32 \times 64$ latitude-longitude grid. We use 29 ocean and atmosphere variables in this study. For ocean variables, we use 1 ocean variables at 9 vertical levels and 1 ocean variables at single level. For atmosphere variables, we use 4 pressure level variables at 4 levels and 3 surface variables. We use the entire 9 CMIP6 datasets from 1850-2014 for pre-training, the 1985-2000 coupled dataset for fine-tuning, the 2001-2004 coupled dataset for validation, and the 2005-2014 coupled dataset for testing. More descriptions about datasets are given in Appendix A.2.1.

**Metrics** Following existing works (Nguyen et al., 2023; Gao et al., 2025), we adopt latitude-weighted Root Mean Square Error (RMSE) and latitude-weighted Anomalous Correlation Coefficient (ACC) to evaluate the forecasting performance of the model, which can be formulated as follows:

$$\text{RMSE}(v) = \frac{1}{HW} \sum_{i,j} L(i) \sqrt{\frac{1}{T} \sum_{t} \left( \hat{X}_{v,i,j}^{t} - X_{v,i,j}^{t} \right)^2},$$

$$\text{ACC}(v) = \frac{1}{HW} \sum_{i,j} L(i) \frac{\sum_t (\hat{X}_{v,i,j}^t - \hat{C}_{v,i,j}^{m_t})(X_{v,i,j}^t - C_{v,i,j}^{m_t})}{\sqrt{\sum_t (\hat{X}_{v,i,j}^t - \hat{C}_{v,i,j}^{m_t})^2 \sum_t (X_{v,i,j}^t - C_{v,i,j}^{m_t})^2}}, \quad (8)$$

where $\text{RMSE}(v)$ and $\text{ACC}(v)$ denote latitude-weighted RMSE and ACC of variable $v$, respectively. $\hat{X}^t, X^t \in \mathbb{R}^{V \times H \times W}$ denote the final forecast and the ground truth at the time $t$, respectively. $\hat{C}^{m_t}, C^{m_t} \in \mathbb{R}^{V \times H \times W}$ denote the forecasting and observed climatology, respectively. The climatology is the long-term mean of the climate state. $m_t$ denotes the month corresponding to the time $t$. $L(i) = H\cos(\phi_i)/\sum_i \cos(\phi_i)$ denotes the latitude weighting factor of the $i$th latitude $\phi_i$. $T$ is the number of test time points.

**Baselines** We compare MMRN with 4 state-of-the-art deep learning models, including OneForecast, Pangu, ClimaX, and GraphCast. Among these baselines, Climax and Pangu are ViT-based models while OneForecast and GraphCast are GNN-based models.

**Implementation details** For the hyperparameter of MMRN, we set the number of scales to 4 and the spatial resolution for each scale is set to (32,64), (16,32), (8,16), (2,4). We set 8 attention heads and 8 Transformer layers for every ViT-based predictor. MMRN is trained/tested on 8 NVIDIA H20 GPUs. AdamW is set as the optimizer with linear warmup and cosine annealing learning rate scheduler. We set the initial learning rate to 7e-4 and 4e-5 for pre-training and fine-tuning, respectively. We set the maximum epochs to 200 for trainging. We employ an early stopping strategy with a tolerance of 5 epochs. We set the batch size to 2 for both process. More descriptions about implementation details are given in Appendix A.2.3.

## 5.1 OVERALL PERFORMANCE

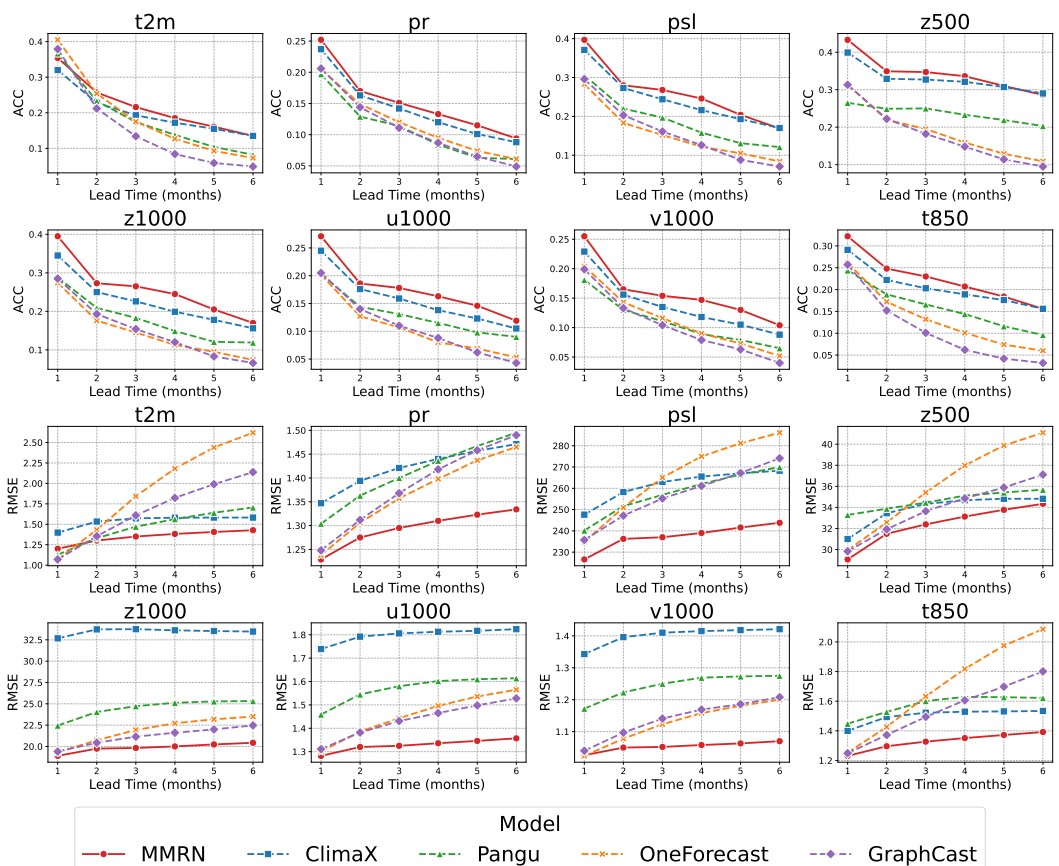

Figure 2: Overall performance of MMRN and baselines in global seasonal climate forecasting.

**Baseline Comparison** We display the performance of MMRN in 8 target variables: 2m temperature (t2m), precipitation (pr), sea level pressure (psl), geopotential at 500hPa (z500), geopotential at 1000hPa (z1000), U wind component at 1000hPa (u1000), V wind component at 1000hPa (v1000), and temperature at 850hPa (t850). Based on the main results shown in Fig. 2, we can observe that: (1) MMRN achieves the best performance in nearly all cases, with an average reduction of 6.04% in RMSE and improvement of 9.12% in ACC compared to the best baseline. We attribute the performance improvement to the design of MMRN, which synergistically combines residual spatial multi-scale modeling with multi-task learning mechanism to effectively capture disentangled multi-scale features and curb hierarchical error propagation. (2) Leveraging the powerful modeling capabilities of ViT, ClimaX effectively models the internal correlations of the climate system and achieves second-best results in most cases. However, ClimaX overlooks the spatial multi-scale characteristic in the climate system and is thereby outperformed by MMRN. (3) Despite adopting spatial multi-scale modeling, the GNN-based models OneForecast and Graphcast exhibit relatively poor performance. This may be that both of them suffer from feature entanglement and hierarchical error propagation in the spatial multi-scale modeling of the climate system.

**Visualization Results** We visualize the spatial distribution of ACC for the temperature at 850hPa (t850) at target lead times of 1, 3, and 6 months in Fig. 3 to analyze the geographical patterns of the model performance. From which we can observe that: (1) All models perform notably better in the low-latitude than high-latitude regions. This may be because the latitude-weighted MSE objective function assigns smaller weights to the high-latitude regions. (2) The ACC of t850 is significantly higher in the eastern equatorial Pacific than over most land regions, which demonstrates that the predictability of t850 is predominantly driven by the oceanic dynamics. More visualizations can be found in Appendix A.3.2.

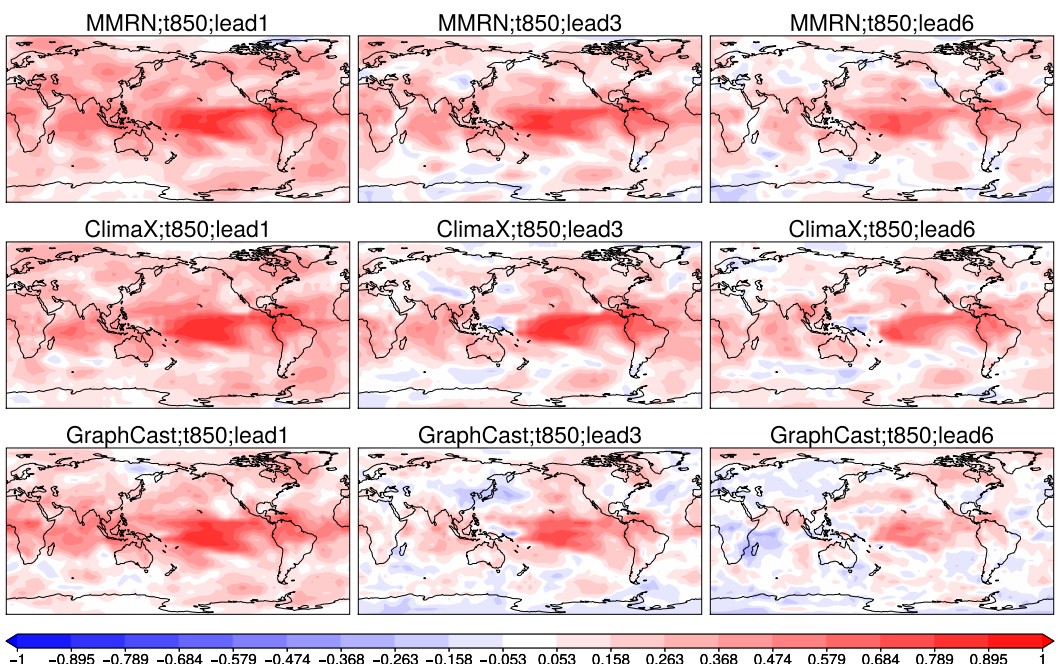

Figure 3: ACC visualization results of t850 at lead time 1, 3, and 6 months.

## 5.2 ZERO-SHOT FORECASTING

We evaluate the generalization ability of MMRN by conducting the zero-shot forecasting experiment. In this experiment, all models are trained only on the 9 datasets from CMIP6 and are evaluated on the coupled dataset without further training. Table 1 presents the averaged results of 4 target variables for lead times ranging from 1 to 6 months. MMRN still achieves the SOTA performance, verifying its powerful generalization ability and knowledge transferability. Specifically, MMRN outperforms the best baseline with an average reduction of 3.55% in RMSE and improvement of 15.26% in ACC, respectively.

Table 1: Zero-shot results in terms of averaged RMSE and ACC. The best results are **bolded** and the second best results are underlined. Full results are listed in Appendix A.3.3.

| | RMSE (↓) | | | | | ACC (↑) | | | | |
| | MMRN | ClimaX | OneForecast | GraphCast | Pangu | MMRN | ClimaX | OneForecast | GraphCast | Pangu |
|------|--------|--------|-------------|-----------|-------|-------|--------|-------------|-----------|-------|
| t2m | **1.572** | 1.750 | 2.362 | 1.767 | 1.754 | **0.222** | 0.167 | 0.165 | 0.182 | 0.164 |
| pr | **1.544E-05** | 1.545E-05 | 1.652E-05 | 1.615E-05 | 1.601E-05 | **0.138** | 0.119 | 0.085 | 0.098 | 0.089 |
| psl | **281.397** | 286.481 | 332.389 | 298.099 | 298.598 | **0.234** | 0.207 | 0.118 | 0.141 | 0.159 |
| z500 | **35.657** | 39.926 | 40.518 | 36.451 | 38.455 | **0.329** | 0.299 | 0.171 | 0.197 | 0.247 |

## 5.3 ABLATION STUDY

To validate the effect of each important design of MMRN on the overall performance, we conduct ablation studies by carefully designing the following 5 variants: (1) Removing the residual connection in the ResMFE module (w/o res). (2) Removing the spatial multi-scale modeling and only forecasting at the original scale (w/o ms). (3) Removing the multi-task learning mechanism in the learning process and only supervising the original scale (w/o mt). (4) Removing the GVE module and embedding ocean and atmosphere variables jointly (w/o gve). (5) Replacing the ResMFE module with a naive small-to-large-scale multi-scale feature extraction method (-r naive mfe). Experimental results from Table 2 show that the lack of any key design will degrade the performance of MMRN, which justifies the effectiveness of our proposed methods. In addition, we can observe

that w/o mt gets the worst performance, indicating the importance of curbing the hierarchical error propagation by imposing supervision at each scale.

Table 2: Ablation results of MMRN. The best results are **bolded**.

| | ACC | | | | | |
|---|---|---|---|---|---|---|
| | MMRN | w/o res | w/o ms | w/o mt | w/o gve | -r naive mfe |
| t2m | **0.218** | 0.205 | 0.195 | 0.203 | 0.200 | 0.206 |
| pr | 0.153 | 0.151 | 0.150 | 0.143 | **0.158** | 0.152 |
| psl | **0.261** | 0.245 | 0.240 | 0.221 | 0.253 | 0.255 |
| z500 | **0.344** | 0.322 | 0.324 | 0.298 | 0.315 | 0.318 |

Table 3: The impact of hyperparameters. The best results are **bolded**

| | Number of Scales | | | | Weight Factor | | | |
|---|---|---|---|---|---|---|---|---|
| | 2 | 3 | 4 | 5 | 0.01 | 0.1 | 0.2 | 0.4 |
| t2m | 0.194 | 0.201 | **0.218** | 0.192 | 0.184 | **0.218** | 0.206 | 0.213 |
| t850 | 0.209 | 0.217 | **0.225** | 0.209 | 0.194 | **0.225** | 0.216 | 0.218 |
| psl | 0.24 | 0.246 | **0.261** | 0.26 | 0.241 | **0.261** | 0.254 | 0.261 |
| zg500 | 0.3 | 0.319 | **0.344** | 0.308 | 0.29 | **0.344** | 0.324 | 0.324 |

## 5.4 HYPERPARAMETER STUDY

We perform hyperparameter studies to evaluate the impact of the number of scales ($S$) and the weight factor of multi-task loss ($\lambda$) in MMRN. For a fair comparison, we fix all other hyperparameters when varying a specific one. Based on the experimental results shown in Table 3, we can observe that: (1) The optimal $S$ is 4, the reason is that a smaller $S$ fails to provide sufficient spatial multi-scale patterns while a larger $S$ may introduce excessive parameters and result in overfitting problems. (2) The optimal $\lambda$ is 0.1. This is because a smaller $\lambda$ provides insufficient supervision for each scale and thus is ineffective at curbing hierarchical error propagation, while a larger $\lambda$ may introduce noise and disrupt the learning of the original scale.

## 5.5 CASE STUDY

The ENSO phenomenon is characterized by an anomalous warming of sea surface temperatures (SST) in the Niño 3.4 region ($170°$W to $190°$E and $5°$S to $5°$N), which is an essential pattern in seasonal climate forecasting. We adopt direct forecasting strategy and assess the ENSO forecasting capability of MMRN by comparing its forecast with observations and other baselines. Fig. 4 demonstrates the performance of MMRN in ENSO forecasting. From which we can observe that: (1) Compared to other baselines, MMRN has higher forecast skill in ENSO forecasting and slower decay of ACC with increasing lead time, achieving ACC of 0.798 at a 6-month lead time. (2) MMRN successfully captures both the pronounced ENSO in 2009-2010 and the moderate one in 2012-2013, demonstrating its robustness of ENSO forecasting.

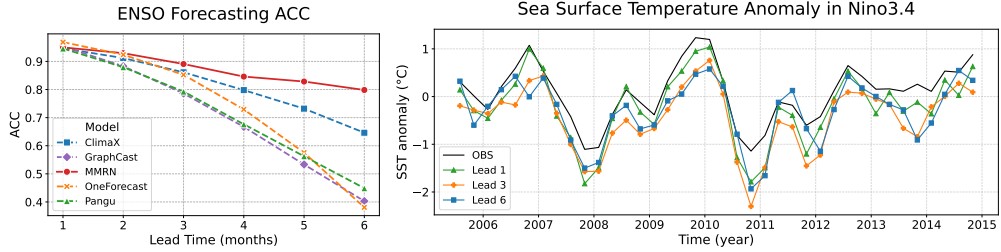

Figure 4: The results of ENSO forecasting. The left figure shows the ACC of ENSO forecasting for lead times ranging from 1 to 6 month. The right figure shows the SST anomaly in Niño 3.4 region from 2005 to 2014.

## 6 CONCLUSIONS

In this paper, we propose MMRN, a first work that applies residual spatial multi-scale modeling and multi-task learning mechanism to the coupled ocean-atmosphere climate system for seasonal climate forecasting. Experimental results demonstrate that MMRN not only achieves the SOTA performance but also has powerful generalization ability. Extensive ablation studies demonstrate the effectiveness of our key designs.

## 7 ETHICS STATEMENT

As our work only focuses on the seasonal climate forecasting problem, there is no potential ethical risk.

## 8 REPRODUCIBILITY STATEMENT

We have rigorously formalized the model architecture, loss functions, and evaluation metrics through illustrations, equations, and descriptions in the main text. All the implementation details, including dataset descriptions, metrics, and experiment configurations, are provided in the manuscript and the code (available online). We provide our source code in an anonymous link: `https://anonymous.4open.science/r/MMRN-4FB5/`, which will be publicly available upon acceptance.

## 9 DECLARATION OF LLM USAGE

The authors use LLM solely as a general-purpose assistive tool for grammar and format refinement. LLM was not used in any of the ideas or technical implementations. The authors take full responsibility for the content of this paper.

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

# A APPENDIX

## A.1 METHOD DETAILS

**Direct Forecasting** A straightforward strategy for seasonal climate forecasting is direct forecasting, which trains a model for each target lead time $T$ to direct forecast the future climate state by $\widehat{X}_{t+T} = \mathcal{F}^T(X_t; \theta_T)$. Direct forecasting can mitigate the error accumulation of auto-regressive forecasting. However, this strategy faces two limitations. Firstly, it is challenging to directly forecast the future climate state since the climate system is chaotic. Secondly, training a model for each target lead time is computationally expensive, especially when the training dataset is large and lead time is long.

**Processing of Ocean Variables** Since ocean variables are undefined over land regions, we replace these undefined values with spatially-shared learnable parameters, which can be formulated as follows:

$$\widetilde{X}_t^{\mathbf{O}} = (1 - M_{land}) \odot X_t^{\mathbf{O}} + M_{land} \odot P_{land}, \tag{9}$$

where $M_{land} \in \mathbb{R}^{V_{\mathbf{O}} \times H \times W}$ denotes a binary mask (1 for land regions, 0 for ocean regions), $\odot$ denotes element-wise multiplication, and $P_{land} \in \mathbb{R}^{V_{\mathbf{O}} \times H \times W}$ denotes trainable parameters shared

across all land regions. $X_t^O, \widetilde{X}_t^O \in \mathbb{R}^{Vo \times H \times W}$ denotes the original ocean variables and processed ocean variables, respectively.

**Pseudocode of the ResMFE**

## A.2 EXPERIMENTAL DETAILS

### A.2.1 DATASET

**CMIP6 Dataset** We use 9 different datasets (i.e., CESM2.r1i1p1f1, CESM2.r2i1p1f1, CESM2.r3i1p1f1, CESM2.r4i1p1f1, CESM2.r5i1p1f1, CESM2.r6i1p1f1, CESM2.r7i1p1f1, CESM2.r8i1p1f1, and CESM2.r10i1p1f1) from the Coupled Model Intercomparison Project Phase 6 (CMIP6) to pretrain all the models. All datasets are historical simulations from the Community Earth System Model version 2 (CESM2) with different initial conditions, developed by the National Center for Atmospheric Research (NCAR). Each dataset consists of 165 years of monthly data, spanning the period from 1850 to 2014. We use all 165 years monthly data for pre-training. All datasets are interpolated to a $5.625° \times 5.625°$ grid by bilinear interpolation.

**ERA5 Dataset** We utilize the atmosphere variables from the ERA5 dataset for fine-tuning, which is the fifth-generation reanalysis product from the European Centre for Medium-Range Weather Forecasts (ECMWF). The ERA5 dataset is characterized by its hourly temporal frequency and global coverage. Spatially, it is provided on a 30 km horizontal grid, with the atmosphere being discretized into 137 vertical levels that span from the surface to an altitude of 80 km. It includes variables for the atmosphere, land, and ocean. Specifically, we selected the data subset spanning from 1958 to 2014 for fine-tuning. The ERA5 dataset is interpolated to a $5.625° \times 5.625°$ grid by bilinear interpolation.

**ORAS5 Dataset** We utilize the ocean variables from the ORAS5 dataset for fine-tuning, which is the reanalysis product of the ECMWF OCEAN5 system. The ORAS5 ocean reanalysis, which extends from 1958 to the present by assimilating in-situ observations of temperature, salinity, and sea ice concentration, served as the data source for our study. Specifically, we selected the data subset spanning from 1958 to 2014 for fine-tuning. The ORAS5 dataset is interpolated to a $5.625° \times 5.625°$ grid by bilinear interpolation.

**Variables** We use 29 ocean and atmosphere variables in this study. For ocean variables, we use 1 ocean variables at 9 vertical levels and 1 ocean variables at single level. For atmosphere variables, we use 4 pressure level variables at 4 levels and 3 surface variables. Table 4 presents the details of these variables.

Table 4: Ocean and atmosphere variables used in this paper.

| Type | Varibale Name | Abbreviation | Levels or Depth |
|---|---|---|---|
| Atmosphere | geopotential | z | 200hPa,500hPa,850hPa,1000hPa |
| Atmosphere | U wind component | u | 200hPa,500hPa,850hPa,1000hPa |
| Atmosphere | V wind component | v | 200hPa,500hPa,850hPa,1000hPa |
| Atmosphere | temperature | t | 200hPa,500hPa,850hPa,1000hPa |
| Atmosphere | 2 metre temperature | t2m | - |
| Atmosphere | precipitation | pr | - |
| Atmosphere | sea level pressure | psl | - |
| Ocean | sea temperature | thetao | 5m,20m,40m,60m,90m,120m,150m,200m,300m |
| Ocean | sea surface height | zos | - |

### A.2.2 BASELINE

**Pangu** Pangu is a Vision Transformer-based deep learning model, which distinguished by two key innovations: a 3D Earth-Specific Transformer (3DEST) that processes Atmosphere data as a unified 3D cube by incorporating different pressure levels, and a hierarchical temporal aggregation algorithm designed to mitigate cumulative errors in long-range predictions.

**GraphCast** GraphCast is a GNN-based weather model that uses an innovative icosahedral grid and an encode-process-decode architecture to efficiently model long-range global weather dynamics.

**Oneforecast** OneForecast is a GNN-based, global-regional nested framework for weather forecasting. It utilizes a multi-scale graph to enable high-resolution regional forecasts and an adaptive messaging mechanism to enhance the prediction of extreme events.

**ClimaX** ClimaX is a Vision Transformer-based foundation model for weather and climate. Its architecture extends the standard Transformer with novel encoding and aggregation blocks to effectively process these varied data inputs.

### A.2.3 IMPLEMENTATION DETAILS

**Baselines** We utilize the source code from `https://github.com/YuanGao-YG/OneForecast` to reproduce Pangu, Oneforecast and GraphCast. We utilize the source code from `https://github.com/microsoft/ClimaX` to reproduce ClimaX. For all the baselines, we all adopt default hyperparameters in the source code to pre-train, fine-tune, and evaluate.

**MMRN** We set the number of scales to 4, the spatial resolution for each scale to (32,64), (16,32), (8,16), (2,4), the patch size of each scale all to (2,2), the dimension of group-wise variables embedder to 128, the weight factor of multi-task learning mechanism to 1 for the original scale and 0.1 for other scales. For each Vision Transformer-based predictor, we set the dimension of predictor to 512, the number of attention heads to 8, the number of Transformer layer to 8, the drop rate of predictor to 0.1, the depth of decoder to 2. In pre-traing stage, we set the batch size to 2 and total batch size is 16, the initial learning rate is 7e-4, the warm up rate of annealing algorithm is 0.04, the max epochs is 200. In fine-tuning stage, we set the batch size to 2 and total batch size is 16, the initial learning rate is 4e-5, the warm up rate of annealing algorithm is 0.01, the max epochs is 200.

### A.3 ADDITIONAL RESULTS

### A.3.1 ADDITIONAL OVERALL PERFORMANCE

In Table 5, we present detailed RMSE and ACC of 18 variables with lead times from 1 to 6 months for the global seasonal climate forecasting. Notably, "var-t" means variable var in lead time t.

Table 5: Detailed global forecasting results of MMRN and baselines. The best results are **bolded**, and the second best results are underlined.

| | RMSE (↓) | | | | | ACC (↑) | | | | |
|---|---|---|---|---|---|---|---|---|---|---|
| | MMRN | ClimaX | Oneforecast | GraphCast | Pangu | MMRN | ClimaX | Oneforecast | GraphCast | Pangu |
| z200-1 | **43.625** | 47.365 | 45.044 | 44.903 | 45.715 | **0.483** | 0.462 | 0.372 | 0.371 | 0.405 |
| z200-2 | 49.267 | 53.090 | 50.749 | 50.193 | **48.319** | **0.389** | 0.371 | 0.262 | 0.256 | 0.332 |
| z200-3 | 51.922 | 55.731 | 56.521 | 53.995 | **49.577** | **0.370** | 0.361 | 0.221 | 0.208 | 0.311 |
| z200-4 | 53.902 | 57.013 | 61.819 | 56.473 | **50.956** | **0.352** | 0.349 | 0.178 | 0.172 | 0.286 |
| z200-5 | 55.422 | 57.478 | 66.249 | 58.549 | **51.828** | 0.331 | **0.332** | 0.147 | 0.137 | 0.273 |
| z200-6 | 56.686 | 57.421 | 69.556 | 61.916 | **52.677** | 0.306 | **0.313** | 0.126 | 0.108 | 0.252 |
| z500-1 | **29.061** | 30.999 | 29.942 | 29.838 | 33.287 | **0.433** | 0.400 | 0.313 | 0.314 | 0.266 |
| z500-2 | **31.497** | 33.452 | 32.570 | 31.922 | 33.878 | **0.349** | 0.330 | 0.221 | 0.222 | 0.250 |
| z500-3 | **32.388** | 34.333 | 35.404 | 33.641 | 34.450 | **0.347** | 0.327 | 0.195 | 0.183 | 0.251 |
| z500-4 | **33.128** | 34.687 | 37.986 | 34.859 | 35.118 | **0.336** | 0.322 | 0.159 | 0.149 | 0.234 |
| z500-5 | **33.771** | 34.821 | 39.879 | 35.890 | 35.417 | **0.310** | 0.308 | 0.130 | 0.114 | 0.219 |
| z500-6 | **34.340** | 34.819 | 41.091 | 37.113 | 35.683 | 0.286 | **0.291** | 0.109 | 0.096 | 0.204 |
| z850-1 | **18.784** | 20.880 | 19.407 | 19.445 | 20.161 | **0.376** | 0.351 | 0.242 | 0.257 | 0.277 |
| z850-2 | **19.686** | 21.696 | 20.480 | 20.325 | 20.685 | **0.268** | 0.266 | 0.155 | 0.178 | 0.212 |
| z850-3 | **19.800** | 21.953 | 21.473 | 20.906 | 21.013 | **0.265** | 0.244 | 0.129 | 0.146 | 0.197 |
| z850-4 | **19.991** | 22.103 | 22.226 | 21.244 | 21.371 | **0.249** | 0.229 | 0.098 | 0.119 | 0.163 |
| z850-5 | **20.216** | 22.217 | 22.658 | 21.552 | 21.617 | **0.215** | 0.211 | 0.077 | 0.079 | 0.138 |
| z850-6 | **20.423** | 22.289 | 22.883 | 21.904 | 21.714 | 0.185 | **0.196** | 0.065 | 0.066 | 0.140 |
| z1000-1 | **18.900** | 32.678 | 19.327 | 19.407 | 22.451 | **0.395** | 0.346 | 0.274 | 0.286 | 0.289 |
| z1000-2 | **19.750** | 33.724 | 20.728 | 20.441 | 24.045 | **0.273** | 0.250 | 0.176 | 0.194 | 0.210 |
| z1000-3 | **19.827** | 33.748 | 21.941 | 21.151 | 24.721 | **0.265** | 0.226 | 0.145 | 0.155 | 0.184 |
| z1000-4 | **20.005** | 33.615 | 22.721 | 21.614 | 25.123 | **0.245** | 0.200 | 0.113 | 0.120 | 0.149 |
| z1000-5 | **20.240** | 33.529 | 23.187 | 21.998 | 25.284 | **0.205** | 0.178 | 0.095 | 0.084 | 0.122 |
| z1000-6 | **20.433** | 33.470 | 23.505 | 22.472 | 25.322 | **0.170** | 0.157 | 0.075 | 0.067 | 0.120 |
| t200-1 | **1.215** | 1.290 | 1.278 | 1.271 | 1.251 | **0.542** | 0.513 | 0.434 | 0.445 | 0.501 |
| t200-2 | **1.388** | 1.449 | 1.423 | 1.402 | 1.391 | **0.427** | 0.397 | 0.313 | 0.328 | 0.389 |
| t200-3 | 1.457 | 1.523 | 1.512 | 1.480 | **1.430** | **0.389** | 0.362 | 0.252 | 0.251 | 0.348 |
| t200-4 | 1.515 | 1.559 | 1.591 | 1.521 | **1.469** | **0.345** | 0.324 | 0.200 | 0.191 | 0.306 |
| t200-5 | 1.559 | 1.573 | 1.657 | 1.540 | **1.491** | **0.311** | 0.297 | 0.159 | 0.162 | 0.273 |

| | | | | | | | | | | |
|---|---|---|---|---|---|---|---|---|---|---|
| t200-6 | 1.589 | 1.578 | 1.708 | 1.587 | **1.507** | **0.283** | 0.271 | 0.143 | 0.133 | 0.247 |
| t500-1 | **1.040** | 1.124 | 1.096 | 1.090 | 1.287 | **0.386** | 0.365 | 0.280 | 0.280 | 0.289 |
| t500-2 | **1.120** | 1.197 | 1.193 | 1.186 | 1.421 | **0.308** | 0.291 | 0.200 | 0.191 | 0.226 |
| t500-3 | **1.165** | 1.241 | 1.303 | 1.265 | 1.480 | **0.288** | 0.279 | 0.170 | 0.150 | 0.220 |
| t500-4 | **1.200** | 1.263 | 1.415 | 1.320 | 1.521 | **0.272** | 0.268 | 0.133 | 0.122 | 0.203 |
| t500-5 | **1.226** | 1.271 | 1.511 | 1.365 | 1.548 | **0.256** | 0.253 | 0.116 | 0.099 | 0.191 |
| t500-6 | **1.249** | 1.272 | 1.581 | 1.435 | 1.566 | 0.235 | **0.241** | 0.109 | 0.083 | 0.181 |
| t850-1 | **1.231** | 1.401 | 1.254 | 1.249 | 1.449 | **0.322** | 0.291 | 0.262 | 0.257 | 0.245 |
| t850-2 | **1.296** | 1.494 | 1.426 | 1.372 | 1.528 | **0.248** | 0.222 | 0.172 | 0.152 | 0.190 |
| t850-3 | **1.327** | 1.523 | 1.632 | 1.495 | 1.601 | **0.230** | 0.203 | 0.133 | 0.102 | 0.167 |
| t850-4 | **1.351** | 1.529 | 1.819 | 1.607 | 1.629 | **0.207** | 0.190 | 0.101 | 0.063 | 0.145 |
| t850-5 | **1.372** | 1.532 | 1.975 | 1.698 | 1.628 | **0.184** | 0.176 | 0.074 | 0.042 | 0.116 |
| t850-6 | **1.392** | 1.534 | 2.086 | 1.801 | 1.622 | 0.156 | **0.157** | 0.061 | 0.032 | 0.097 |
| u200-1 | **4.365** | 4.708 | 4.679 | 4.667 | 4.644 | **0.396** | 0.375 | 0.266 | 0.279 | 0.327 |
| u200-2 | **4.591** | 5.007 | 4.900 | 4.850 | 4.812 | **0.278** | 0.266 | 0.179 | 0.194 | 0.242 |
| u200-3 | **4.635** | 5.144 | 5.034 | 5.000 | 4.876 | **0.258** | 0.241 | 0.153 | 0.152 | 0.225 |
| u200-4 | **4.693** | 5.227 | 5.160 | 5.098 | 4.962 | **0.232** | 0.221 | 0.127 | 0.124 | 0.195 |
| u200-5 | **4.743** | 5.298 | 5.242 | 5.125 | 5.053 | **0.205** | 0.185 | 0.114 | 0.111 | 0.159 |
| u200-6 | **4.806** | 5.365 | 5.311 | 5.231 | 5.107 | **0.168** | 0.154 | 0.098 | 0.092 | 0.141 |
| u500-1 | **2.899** | 3.070 | 3.025 | 3.017 | 3.047 | **0.309** | 0.294 | 0.197 | 0.212 | 0.244 |
| u500-2 | **3.006** | 3.213 | 3.135 | 3.101 | 3.140 | **0.195** | 0.186 | 0.120 | 0.139 | 0.165 |
| u500-3 | **3.014** | 3.259 | 3.204 | 3.162 | 3.173 | **0.184** | 0.167 | 0.108 | 0.112 | 0.157 |
| u500-4 | **3.031** | 3.278 | 3.269 | 3.197 | 3.211 | **0.176** | 0.158 | 0.092 | 0.093 | 0.139 |
| u500-5 | **3.049** | 3.293 | 3.311 | 3.220 | 3.248 | **0.159** | 0.139 | 0.079 | 0.073 | 0.113 |
| u500-6 | **3.073** | 3.311 | 3.343 | 3.257 | 3.272 | **0.132** | 0.120 | 0.064 | 0.059 | 0.094 |
| u850-1 | **1.805** | 1.974 | 1.851 | 1.852 | 1.915 | **0.298** | 0.289 | 0.207 | 0.217 | 0.231 |
| u850-2 | **1.866** | 2.048 | 1.951 | 1.927 | 1.963 | **0.202** | 0.200 | 0.130 | 0.148 | 0.156 |
| u850-3 | **1.872** | 2.088 | 2.018 | 1.979 | 1.982 | **0.200** | 0.183 | 0.115 | 0.122 | 0.150 |
| u850-4 | **1.887** | 2.111 | 2.078 | 2.018 | 2.008 | **0.183** | 0.166 | 0.090 | 0.098 | 0.132 |
| u850-5 | **1.901** | 2.127 | 2.121 | 2.058 | 2.026 | **0.163** | 0.151 | 0.074 | 0.069 | 0.112 |
| u850-6 | **1.915** | 2.140 | 2.150 | 2.098 | 2.034 | **0.138** | 0.133 | 0.060 | 0.050 | 0.105 |
| u1000-1 | **1.282** | 1.739 | 1.302 | 1.312 | 1.459 | **0.271** | 0.245 | 0.202 | 0.205 | 0.204 |
| u1000-2 | **1.320** | 1.792 | 1.387 | 1.383 | 1.546 | **0.186** | 0.177 | 0.128 | 0.140 | 0.144 |
| u1000-3 | **1.325** | 1.807 | 1.444 | 1.431 | 1.581 | **0.178** | 0.160 | 0.107 | 0.110 | 0.132 |
| u1000-4 | **1.336** | 1.813 | 1.496 | 1.466 | 1.602 | **0.163** | 0.138 | 0.081 | 0.089 | 0.115 |
| u1000-5 | **1.346** | 1.818 | 1.536 | 1.498 | 1.610 | **0.146** | 0.123 | 0.069 | 0.063 | 0.099 |
| u1000-6 | **1.357** | 1.824 | 1.566 | 1.529 | 1.615 | **0.119** | 0.105 | 0.053 | 0.043 | 0.090 |
| v200-1 | **3.317** | 3.453 | 3.380 | 3.400 | 3.466 | **0.250** | 0.232 | 0.175 | 0.167 | 0.192 |
| v200-2 | **3.382** | 3.515 | 3.417 | 3.438 | 3.485 | **0.164** | 0.159 | 0.134 | 0.122 | 0.129 |
| v200-3 | **3.381** | 3.544 | 3.454 | 3.471 | 3.486 | **0.161** | 0.150 | 0.114 | 0.098 | 0.125 |
| v200-4 | **3.393** | 3.562 | 3.492 | 3.497 | 3.527 | **0.151** | 0.143 | 0.098 | 0.080 | 0.095 |
| v200-5 | **3.401** | 3.577 | 3.518 | 3.514 | 3.546 | **0.134** | 0.124 | 0.079 | 0.070 | 0.072 |
| v200-6 | **3.424** | 3.597 | 3.544 | 3.552 | 3.566 | 0.107 | **0.109** | 0.057 | 0.052 | 0.063 |
| v500-1 | **2.046** | 2.089 | 2.063 | 2.073 | 2.131 | **0.178** | 0.174 | 0.118 | 0.115 | 0.134 |
| v500-2 | **2.080** | 2.127 | 2.089 | 2.089 | 2.130 | 0.098 | **0.102** | 0.082 | 0.080 | 0.075 |
| v500-3 | **2.071** | 2.137 | 2.109 | 2.105 | 2.126 | **0.102** | 0.100 | 0.066 | 0.064 | 0.079 |
| v500-4 | **2.073** | 2.143 | 2.128 | 2.123 | 2.145 | **0.100** | 0.093 | 0.061 | 0.054 | 0.061 |
| v500-5 | **2.076** | 2.146 | 2.139 | 2.135 | 2.152 | **0.089** | 0.086 | 0.044 | 0.042 | 0.042 |
| v500-6 | **2.087** | 2.152 | 2.153 | 2.154 | 2.158 | 0.065 | **0.076** | 0.029 | 0.026 | 0.038 |
| v850-1 | **1.272** | 1.391 | 1.279 | 1.289 | 1.364 | **0.229** | 0.224 | 0.169 | 0.167 | 0.162 |
| v850-2 | **1.294** | 1.432 | 1.314 | 1.321 | 1.376 | **0.154** | 0.149 | 0.125 | 0.116 | 0.121 |
| v850-3 | **1.293** | 1.443 | 1.346 | 1.348 | 1.388 | **0.147** | 0.132 | 0.101 | 0.096 | 0.110 |
| v850-4 | **1.297** | 1.447 | 1.374 | 1.370 | 1.403 | **0.141** | 0.120 | 0.083 | 0.079 | 0.087 |
| v850-5 | **1.301** | 1.448 | 1.393 | 1.386 | 1.407 | **0.123** | 0.109 | 0.068 | 0.061 | 0.072 |
| v850-6 | **1.308** | 1.452 | 1.409 | 1.411 | 1.409 | 0.096 | **0.098** | 0.050 | 0.039 | 0.063 |
| v1000-1 | 1.026 | 1.343 | **1.025** | 1.040 | 1.173 | **0.255** | 0.230 | 0.204 | 0.199 | 0.181 |
| v1000-2 | **1.050** | 1.397 | 1.079 | 1.098 | 1.223 | **0.165** | 0.156 | 0.144 | 0.133 | 0.130 |
| v1000-3 | **1.052** | 1.411 | 1.124 | 1.142 | 1.251 | **0.154** | 0.136 | 0.116 | 0.105 | 0.111 |
| v1000-4 | **1.058** | 1.416 | 1.159 | 1.169 | 1.269 | **0.147** | 0.118 | 0.091 | 0.079 | 0.090 |
| v1000-5 | **1.063** | 1.418 | 1.181 | 1.186 | 1.273 | **0.130** | 0.105 | 0.073 | 0.063 | 0.079 |
| v1000-6 | **1.070** | 1.422 | 1.201 | 1.209 | 1.275 | **0.104** | 0.089 | 0.053 | 0.040 | 0.065 |
| t2m-1 | 1.201 | 1.399 | 1.071 | **1.070** | 1.128 | 0.354 | 0.321 | **0.405** | 0.380 | 0.368 |
| t2m-2 | **1.301** | 1.534 | 1.432 | 1.354 | 1.330 | **0.257** | 0.228 | 0.255 | 0.213 | 0.234 |
| t2m-3 | **1.350** | 1.574 | 1.843 | 1.609 | 1.470 | **0.216** | 0.194 | 0.175 | 0.135 | 0.175 |
| t2m-4 | **1.382** | 1.580 | 2.182 | 1.824 | 1.563 | **0.185** | 0.173 | 0.128 | 0.085 | 0.139 |
| t2m-5 | **1.406** | 1.582 | 2.441 | 1.990 | 1.641 | **0.161** | 0.155 | 0.094 | 0.060 | 0.105 |
| t2m-6 | **1.427** | 1.583 | 2.620 | 2.139 | 1.707 | **0.135** | 0.135 | 0.073 | 0.049 | 0.083 |
| pr-1 | **1.229E-05** | 1.348E-05 | 1.238E-05 | 1.248E-05 | 1.304E-05 | **0.252** | 0.237 | 0.206 | 0.207 | 0.198 |
| pr-2 | **1.275E-05** | 1.394E-05 | 1.306E-05 | 1.312E-05 | 1.364E-05 | **0.170** | 0.163 | 0.150 | 0.145 | 0.129 |
| pr-3 | **1.295E-05** | 1.421E-05 | 1.358E-05 | 1.369E-05 | 1.400E-05 | **0.151** | 0.143 | 0.120 | 0.112 | 0.115 |
| pr-4 | **1.310E-05** | 1.441E-05 | 1.399E-05 | 1.419E-05 | 1.437E-05 | **0.133** | 0.121 | 0.096 | 0.087 | 0.085 |
| pr-5 | **1.323E-05** | 1.457E-05 | 1.437E-05 | 1.458E-05 | 1.468E-05 | **0.115** | 0.101 | 0.074 | 0.065 | 0.064 |

| | | | | | | | | | |
|---|---|---|---|---|---|---|---|---|---|
| pr-6 | **1.334E-05** | 1.471E-05 | 1.465E-05 | 1.490E-05 | 1.496E-05 | **0.094** | 0.088 | 0.061 | 0.050 | 0.061 |
| psl-1 | **226.589** | 247.614 | 234.854 | 235.734 | 240.178 | **0.397** | 0.371 | 0.284 | 0.296 | 0.306 |
| psl-2 | **236.244** | 258.259 | 250.948 | 247.151 | 251.623 | **0.280** | 0.273 | 0.183 | 0.203 | 0.220 |
| psl-3 | **237.001** | 263.005 | 265.080 | 255.226 | 256.901 | **0.268** | 0.244 | 0.152 | 0.162 | 0.197 |
| psl-4 | **239.008** | 265.500 | 274.993 | 261.111 | 262.041 | **0.246** | 0.216 | 0.122 | 0.127 | 0.158 |
| psl-5 | **241.563** | 267.052 | 281.166 | 267.185 | 266.494 | **0.204** | 0.194 | 0.106 | 0.089 | 0.131 |

### A.3.2 ADDITIONAL VISUALIZATIONS

We visualize the RMSE distribution of t850 in the Fig. 5. From which we can observe that the spatial distribution of RMSE exhibits a pattern consistent with that of ACC, indicating higher forecast accuracy at lower latitudes compared to higher latitudes, and over oceans compared to land. In addition, we visualize the ACC distribution of u1000, v1000, and z1000 in the Fig. 6, Fig. 7, and Fig. 8, respectively. From which we can observe that MMRN demonstrates a slower decay of ACC with increasing lead time. We attribute this to our novel multi-scale modeling framework, which can learn disentangled multi-scale features and curb the hierarchical propagation.

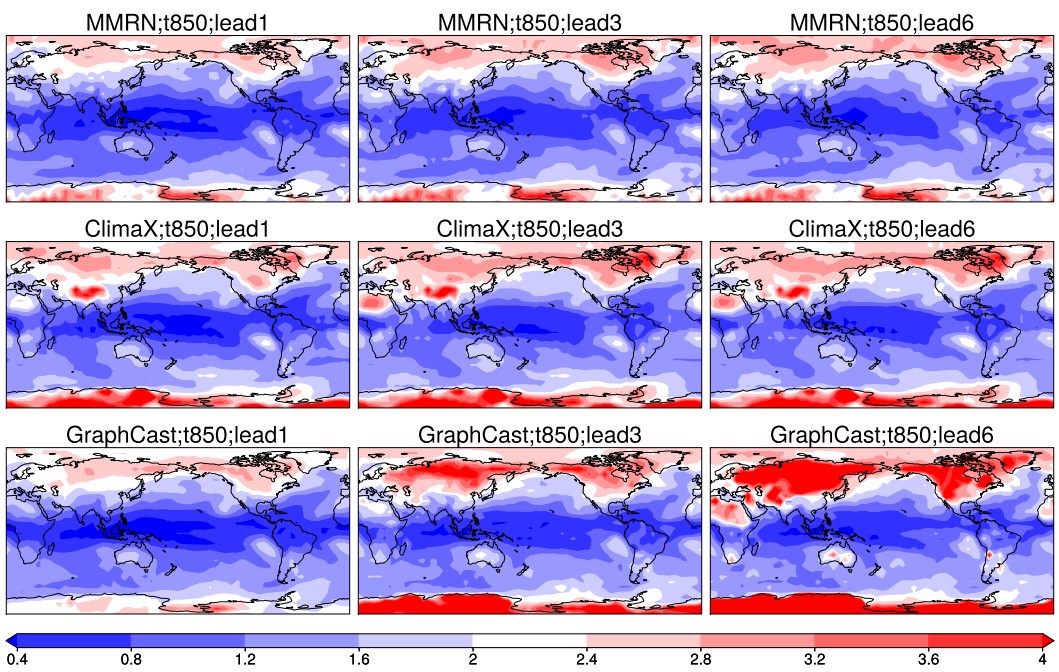

Figure 5: RMSE visualization results of t850 at lead time 1, 3, and 6 months.

### A.3.3 ADDITIONAL ZERO-SHOT FORECASTING

Table 6 present detailed zero-shot forecasting results of 4 variables with lead times from 1 to 6 month in RMSE and ACC.

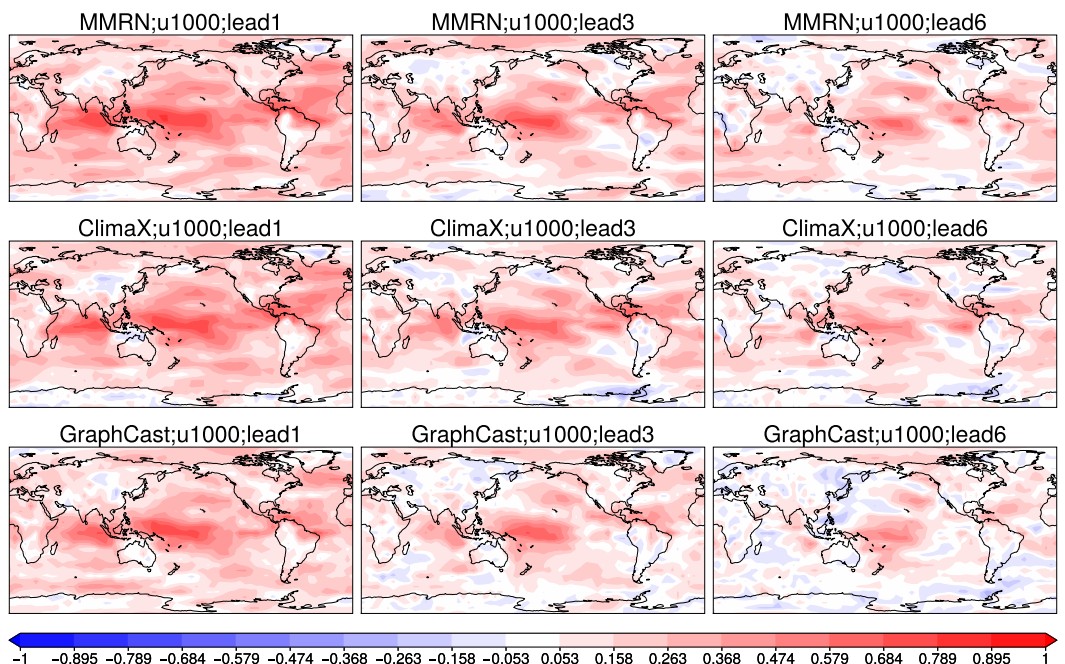

Figure 6: ACC visualization results of u1000 at lead time 1, 3, and 6 months.

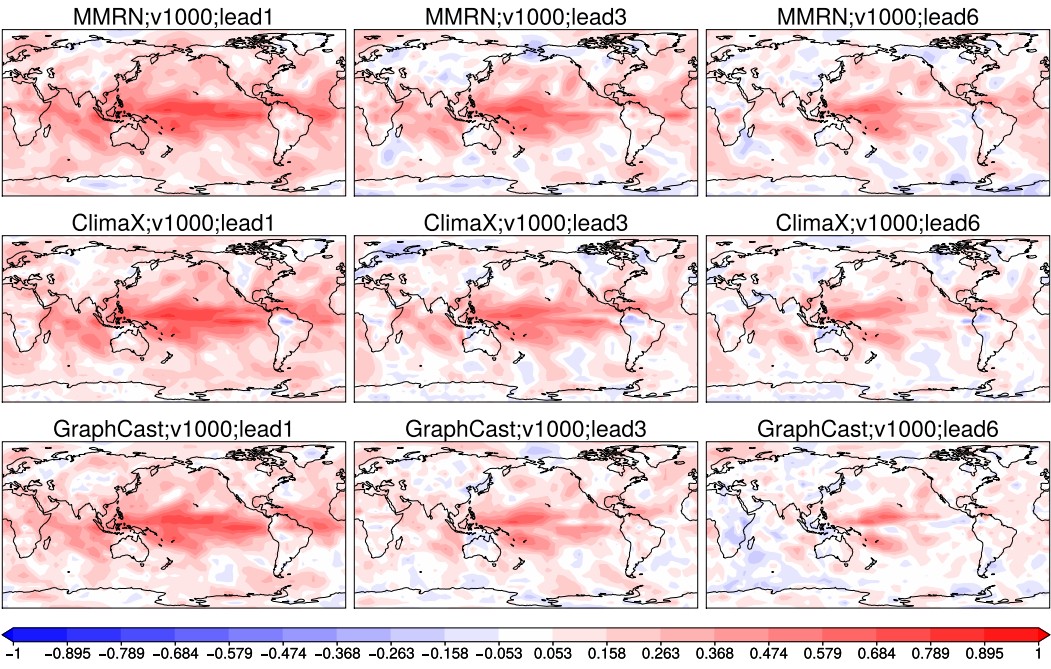

Figure 7: ACC visualization results of v1000 at lead time 1, 3, and 6 months.

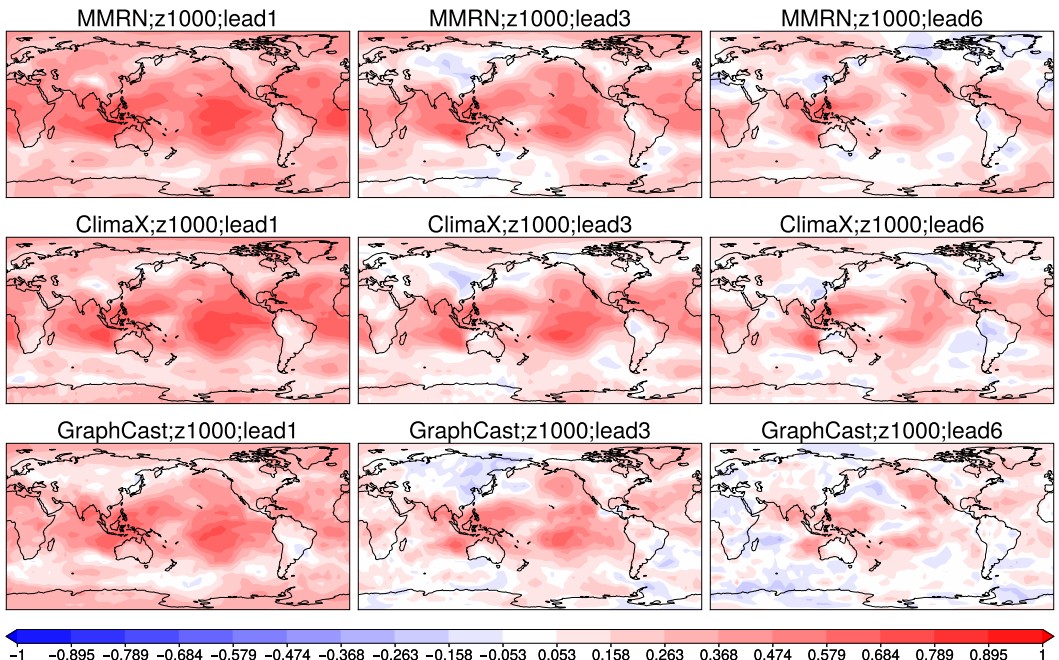

Figure 8: ACC visualization results of z1000 at lead time 1, 3, and 6 months.

Table 6: Detailed zero-shot forecasting results of MMRN and baselines. The best results are **bolded**, and the second best results are underlined.

| | | | RMSE | | | | | ACC | | |
|---|---|---|---|---|---|---|---|---|---|---|
| | MMRN | ClimaX | Oneforecast | GraphCast | Pangu | MMRN | ClimaX | Oneforecast | GraphCast | Pangu |
| t2m-1 | 1.489 | 1.582 | 1.403 | **1.233** | 1.327 | 0.364 | 0.321 | **0.402** | 0.401 | 0.343 |
| t2m-2 | 1.555 | 1.723 | 1.886 | **1.529** | 1.613 | **0.263** | 0.208 | 0.245 | 0.251 | 0.210 |
| t2m-3 | **1.575** | 1.809 | 2.324 | 1.747 | 1.778 | **0.215** | 0.149 | 0.152 | 0.170 | 0.156 |
| t2m-4 | **1.590** | 1.822 | 2.661 | 1.912 | 1.876 | **0.185** | 0.093 | 0.125 | 0.122 | 0.122 |
| t2m-5 | **1.606** | 1.801 | 2.887 | 2.039 | 1.941 | **0.160** | 0.107 | 0.060 | 0.090 | 0.090 |
| t2m-6 | **1.620** | 1.766 | 3.011 | 2.140 | 1.991 | **0.143** | 0.092 | 0.045 | 0.060 | 0.063 |
| pr-1 | 0.142E-05 | **0.141E-05** | 0.144E-05 | 0.147E-05 | 0.146E-05 | **0.232** | 0.232 | 0.183 | 0.190 | 0.185 |
| pr-2 | 0.150E-05 | **0.150E-05** | 0.156E-05 | 0.156 E-05 | 0.156E-05 | **0.159** | 0.143 | 0.118 | 0.128 | 0.112 |
| pr-3 | **0.155E-05** | 0.155E-05 | 0.164E-05 | 0.161E-05 | 0.160E-05 | **0.136** | 0.114 | 0.084 | 0.102 | 0.092 |
| pr-4 | **0.158E-05** | 0.159E-05 | 0.171E-05 | 0.165E-05 | 0.163E-05 | **0.113** | 0.091 | 0.057 | 0.079 | 0.070 |
| pr-5 | **0.160E-05** | 0.161E-05 | 0.176E-05 | 0.168E-05 | 0.167E-05 | **0.097** | 0.071 | 0.039 | 0.058 | 0.044 |
| pr-6 | **0.161E-05** | 0.162E-05 | 0.180E-05 | 0.171E-05 | 0.170E-05 | **0.090** | 0.067 | 0.030 | 0.032 | 0.035 |
| psl-1 | 261.144 | **261.079** | 290.424 | 275.597 | 274.432 | **0.380** | 0.371 | 0.263 | 0.280 | 0.284 |
| psl-2 | 281.960 | **280.221** | 315.811 | 290.070 | 297.185 | **0.268** | 0.248 | 0.168 | 0.190 | 0.196 |
| psl-3 | **285.145** | 289.461 | 333.524 | 297.733 | 300.327 | **0.230** | 0.195 | 0.110 | 0.148 | 0.158 |
| psl-4 | **285.766** | 295.942 | 345.122 | 303.517 | 303.572 | **0.200** | 0.152 | 0.085 | 0.113 | 0.129 |
| psl-5 | **287.076** | 296.579 | 352.902 | 308.908 | 307.213 | **0.169** | 0.141 | 0.050 | 0.069 | 0.110 |
| psl-6 | **287.295** | 295.606 | 356.553 | 312.769 | 308.861 | **0.154** | 0.138 | 0.034 | 0.047 | 0.083 |
| z500-1 | 34.861 | 36.659 | **34.510** | 34.795 | 37.038 | **0.422** | 0.410 | 0.295 | 0.309 | 0.309 |
| z500-2 | **35.784** | 39.313 | 37.715 | 36.049 | 39.055 | **0.346** | 0.313 | 0.210 | 0.233 | 0.261 |
| z500-3 | **35.712** | 40.974 | 40.164 | 36.469 | 38.708 | **0.330** | 0.282 | 0.171 | 0.206 | 0.257 |
| z500-4 | **35.591** | 41.077 | 42.345 | 36.788 | 38.527 | **0.315** | 0.276 | 0.143 | 0.177 | 0.243 |
| z500-5 | **35.825** | 40.940 | 43.858 | 37.137 | 38.696 | **0.287** | 0.260 | 0.120 | 0.145 | 0.219 |
| z500-6 | **36.171** | 40.590 | 44.513 | 37.468 | 38.704 | **0.275** | 0.258 | 0.087 | 0.114 | 0.197 |

