# OpenReview forum: "MMRN: A Multi-scale Multi-task Residual Network for Seasonal Climate Forecasting"
_ICLR.cc/2026/Conference — Submitted to ICLR 2026_

### Official Review · Reviewer_HeCm · 2025-10-27

**Soundness:** 2
**Presentation:** 3
**Contribution:** 2
**Rating:** 2
**Confidence:** 4

**Summary:**

This paper introduces the Multi-scale Multi-task Residual Network (MMRN) for seasonal climate forecasting in the coupled ocean-atmosphere system. The model tackles two central challenges in multi-scale climate modeling: feature entanglement across scales and variables, and hierarchical error propagation. MMRN leverages a group-wise variable embedder, a residual multi-scale feature extractor, and applies a multi-task learning scheme with scale-specific supervision to explicitly disentangle features and curb error accumulation. Extensive experiments on global reanalysis datasets show state-of-the-art results over strong baselines, with detailed ablations and visualizations supporting the claims.

**Strengths:**

Architectural Clarity: Figure 1 gives a transparent, modular view of the architecture, clearly demarcating the role of the GVE, ResMFE, ViT predictors, aggregation, and multi-task supervision. This supports reader understanding and reproducibility.

Sound Mathematical Formulation: The residual update strategy in the ResMFE module and scale-specific losses are derived with explicit equations, offering clarity on optimization and training mechanics.

Reproducibility Commitment: Source code and implementation details are promised and methodologically described, including open-source code (link provided) and specifics on datasets, baselines, and environment.

**Weaknesses:**

1. In the Introduction section, the authors devote an excessive amount of space to describing *seasonal climate forecasting*, while the actual core theme of the paper is seasonal climate forecasting in the coupled ocean–atmosphere system. This core focus is only mentioned in the latter part of the Introduction, making it difficult for readers to grasp the paper’s motivation and research objective.
2. The authors’ motivation is not clearly articulated. They propose two challenges—*Feature Entanglement* and *Hierarchical Error Propagation*. However, in the discussion of Feature Entanglement, the authors first emphasize the issue of scale, which could easily be categorized under Hierarchical Error Propagation.
3. In the Introduction, the problems identified are all discussed in the context of general *seasonal climate forecasting*, whereas the central topic of the paper specifically concerns the *coupled ocean–atmosphere system*. This lack of alignment undermines the thematic consistency of the paper.
4. In the Introduction, the *multi-task learning mechanism* appears abruptly without any conceptual buildup or motivation. The introduction of this mechanism is unclear. Moreover, claiming that defining separate predictions for different scales constitutes multi-task learning is an overstatement.
5. At line 120, the authors define the variable *X*, but they redefine *X* again at line 188, using different variable symbols (*C* and *V*). This results in a redundant and inconsistent definition of variables.
6. Although the multi-task learning setup (Eqn. for $\mathcal{L}_{MT}$) is well-motivated, the effect of different weighting strategies ($\lambda_s$) is checked only empirically (Table 3). There is no principled guidance or a sensitivity analysis in varied data regimes (e.g., if S is larger or if data is more spatially heterogeneous). As spectral bias and error propagation are fundamental to the paper’s argument, the absence of deeper analysis here is a missed opportunity for conceptual impact.

**Questions:**

The same as weaknesses.

---

### Official Review · Reviewer_VENE · 2025-10-27

**Soundness:** 2
**Presentation:** 2
**Contribution:** 2
**Rating:** 2
**Confidence:** 5

**Summary:**

The paper introduces a Multi-scale Multi-task Residual Network (MMRN) for seasonal climate forecasting, addressing predictions 3-6 months ahead using coupled ocean-atmosphere data. It first uses a group-wise variable embedder to independently embed ocean and atmosphere variables, mitigating cross-variable entanglement, followed by parallel residual multi-scale feature extractors that iteratively extract disentangled features from the different scales. These features are then fed into ViT-based predictors for scale-specific forecasting, aggregated hierarchically to form the final prediction. The model is pre-trained on CMIP6 datasets and fine-tuned on aligned ERA5/ORAS5 reanalysis data.

**Strengths:**

The authors propose to address vital seasonal forecasting while identifying two key technical issues. Specifically,

* The authors consider interactions across different spatial scales.
* The paper emphasizes the interaction between atmospheric and oceanic variables.
* The use of CMIP6 data for pre-training the model and the inclusion of ocean data at various vertical levels, enriching the dataset's diversity and depth.
* The authors test the model performance of an important ENSO prediction.

**Weaknesses:**

**Method**
--------------------------------

After carefully reviewing the paper,  I found that the proposed method is specialized to the considered resolution of weather data. The 5.625° resolution is generally too coarse to represent local weather dynamics. Thus, the claimed multi-scale framework is not convincing since the resolutions are all coarse. Instead of incorporating these techniques, an empirical solution is to increase the data resolution, which is totally feasible. As a reference, the spatial resolutions of NeuralGCM are 2.8°, 1.4°, and 0.7°. The remaining technical issues are as follows:

* Given the already coarse spatial resolution of the input data, the model could sufficiently operate at the original scale without introducing multi-scale processing, raising questions about the necessity of downsampling to even coarser resolutions followed by upsampling.
* Equation 4, which defines the iterative residual feature extraction via downsampling and upsampling, lacks clear motivation. Why this specific formulation is superior to directly applying a standard CNN or other feature extraction approaches remains unexplained.
* In line 62, how do you define the model’s ability to disentangle and learn multiscale features? The reference is not related.
* Writing issues, including overly verbose and non-concise mathematical notations in the method section, the absence of mathematical formulas to describe the ViT-based Predictor (relying instead on textual explanations), and the creation of unnecessary concepts that complicate understanding, for example, the Group-wise Variable Embedder (GVE) module appears to simply partition and embed ocean and atmosphere variables independently using separate linear layers.

Overall, the MMRN model designs lack deep insights into the underlying mechanisms of seasonal climate forecasting.

**Results**
-------------------------------

For experiments, I have the following questions:
* The authors do not provide sufficient specifics on how baselines such as GraphCast and PanguWeather are adapted to the paper's coarser resolution. Critical adaptations, including how the icosahedral mesh is constructed or coarsened for GraphCast's graph neural network message passing, and how the input grid is tokenized into patches for PanguWeather's Vision Transformer, are entirely omitted.
* OneForecast, Pangu, ClimaX, and GraphCast are not initially designed for seasonal forecasting. An important baseline is NeuralGCM. The authors miss the comparison as well as the citation.
* The authors do not compare the model performance against a standard climatology baseline. The climatology method serves as a simple "no-skill" benchmark, where predictions for a given month or season are based on the long-term historical average (typically computed over a 30-year baseline period) of meteorological variables from reanalysis or observational data. This baseline is important because it represents the fundamental level of predictability inherent in the climate system's seasonal patterns; any advanced model must outperform it to demonstrate genuine skill.

**Questions:**

* Could the model operate effectively on the high-resolution data?
* What are the insights behind the model designs?
* Please provide the implementation details of baselines.
* Why was NeuralGCM, a more relevant baseline for this task, omitted from the comparison, and is not cited?
* What is the performance of the climatology baseline?

**Details Of Ethics Concerns:**

N/A.

---

### Official Review · Reviewer_rEpq · 2025-10-27

**Soundness:** 4
**Presentation:** 4
**Contribution:** 3
**Rating:** 8
**Confidence:** 4

**Summary:**

The authors propose a new method for seasonal climate forecasting of weather variables. The authors propose two key innovations compared to existing methods: split the embedding of variables into groups (Atmospheric variables and Ocean variables) (Group-wise Variable Embedder (GVE) Module) and explicitly extract features at varying spatial resolutions (Residual Multi-scale Feature Extractor (ResMFE) Module).
These two modules are proposed to overcome two issues that the authors identified: 1) Feature entanglement and 2) Hierarchical Error Propagation.
GVE simply embeds the two groups separately.
ResMFE on the other hand is a bit more involved. At each scale, the downscaled feature map is upscaled and subtracted from the original feature map to extract scale specific features.
ResMFE is applied separately to each group, and then merged in a scale aware manner (Disentangled Multi-scale Features and Hierarchical Aggregation Module).
Save for a couple lead times and variables, the proposed method seem to mostly outperform existing state of the art methods. An extensive ablation study consolidates the findings and asserts the importance of each module.

**Strengths:**

The paper is well written and easy to follow, the proposed methods are interesting and the figures well made.
The proposed method does beat state of the art models in the field, and seems to add to the literature.
I am particularly drown to the ResMFE method, which I think is smart and I haven't seen before. This is akin to a reverse skip connection method: instead of adding information to the downscaled feature map, the feature map is upscaled and removes information from the original map. This seems very interesting and could have implications for other environmental applications, where multi-scale processes are highly relevant.

**Weaknesses:**

I am less convinced by the GVE module. The ablation study even shows that for certain variables this is harmful, and overall it does not appear to be improving the results that much. It would be good to have table 2 in percent of improvement over baseline in order to more easily compare the methods.
I understand the logic and reasoning behind GVE, and I do think the feature entanglement is something to be worry of, but I do not think the proposed method is an answer to that.
Splitting the variables into those two groups seems somewhat arbitrary, and more driven by the data source than other logics. The remaining features in each group are probably as entangled as the features between groups would be.
I think this weakens the paper a bit, as the reasoning for GVE isn't reflected in the execution (although the paper remains very good). I would like to see a further study on different ways to disentangle the variables, e.g. by encoding each variable separately, or by grouping them in a different manner.

Minor issues:
- Line 014: "two key challenges limit its further development:" missing 's (its further development)
- Line 133-148: this feels like a repetition of the experiment section
- Figure 1: after EtA, ResMFE should be shown as a module
- Figure 1: the caption should explain the figure
- Figure 2, 3, 4, etc: same, caption is incomplete
- Line 254: "multi-task" this feels more like "Multi-Scale" rather than "Multi-task". I would consider changing it.

**Questions:**

I'd like to see more work on the variables disentanglement.

---

### Official Review · Reviewer_uy1R · 2025-10-30

**Soundness:** 2
**Presentation:** 2
**Contribution:** 1
**Rating:** 2
**Confidence:** 5

**Summary:**

This paper proposes a multi-scale network for seasonal climate forecasting. The author aims to learning different scales of atmosphere and ocean.  The authors identify two major challenges in existing deep learning methods: feature entanglement across scales and variables and hierarchical error propagation. The results seems better than baselines, however, I have many concerns of this work.

**Strengths:**

1. MMRN consistently outperforms advanced baselines across lead times and variables.
2. The writing is clear.

**Weaknesses:**

1. The definition of the title is inaccurate.  The seasonal forecasting is not within the scope of climate prediction. Climate forecasting targets for multi-year prediction.
2. This work lacks novelty. From my view, the proposed architecture is a simple combination of some common methods. For instance, ResMFE is similar to [1]. And this paper doesn’t demonstrate ResMFE can learning different scales of physical fields. The authors does not clearly define what a multi-scale physical field is. Although simple upsampling and downsampling can theoretically extract features from different scales, the authors don’t provide a clear analysis. And Multi-task Learning Mechanism is very simple in computer vision. Further, the so called coupled ocean-atmosphere model is just the the stacking of variables.

3.  The data used in this work is too coarse. 5.6 degree (32✖️64) only composes large-scale features, and lacks details (corresponding to small-scale features). If the authors intent to improve this work, the resolution is at least 0.25 degree (721✖️1440), a common resolution used by previous work, such as Fengwu, Fuxi, Graphcast, and Pangu.

4. The baselines are unfair. OneForecast, Pangu, and Graphcast are designed for medium-range weather forecasting, rather than long-term forecasting. For long-term forecasting, CirT [2], ORCA_DL [3], Fuxi-S2S [4]should be added. Further, a coupled ocean-atmosphere model [5] should be added for comparison.

5. The overall architecture is a dual-branch network. Modules such as GVE may not have played a role in improving the paper. Although the ablation experiments gave some results, this is likely due to the increase in the number of parameters and certain layers. Other structures can achieve the same effect.



[1] Helmfluid: Learning helmholtz dynamics for interpretable fluid prediction

[2] CirT: Global Subseasonal-to-Seasonal Forecasting with Geometry-inspired Transformer

[3] Data-driven global ocean modeling for seasonal to decadal prediction

[4] A machine learning model that outperforms conventional global subseasonal forecast models

[5] Coupled ocean-atmosphere dynamics in a machine learning earth system model

**Questions:**

See weakness.

---

### Meta-Review · Area_Chair_NV6s · 2025-12-05

**Summary:**

This paper received an initial score of 2/2/2/8, i.e., 3 out of 4 reviewers recommended strong rejection. Considering that the authors did not provide a rebuttal for the paper, the AC decided to follow the majority of reviewers to recommend rejection. The AC agreed with the reviewers on some major weaknesses of the paper, especially the very low-resolution forecasts and the improper baselines, based on which the paper is not yet ready for publication.

**Reviewer Concerns:**

No rebuttal.

**Reviewer Scores:**

No rebuttal. I think the reviewers would keep their scores unchanged.

---

### Decision · Program_Chairs · 2026-01-26

Reject